# Age and micronutrient effects on the microbiome in a mouse model of zinc depletion and supplementation

Edward W. Davis, II[1], Carmen P. Wong[2,3], Holly K. Arnold[4], Kristin Kasschau[4], Christopher A. Gaulke[4¤], Thomas J. Sharpton[4,5], Emily Ho[2,3]*

1 Center for Quantitative Life Sciences, Oregon State University, Corvallis, Oregon, United States of America, 2 Linus Pauling Institute, Oregon State University, Corvallis, Oregon, United States of America, 3 School of Biological and Population Health Sciences, Oregon State University, Corvallis, Oregon, United States of America, 4 Department of Microbiology, Oregon State University, Corvallis, Oregon, United States of America, 5 Department of Statistics, Oregon State University, Corvallis, Oregon, United States of America

¤ Current address: Department of Pathobiology, University of Illinois Urbana-Champaign, Urbana, Illinois, United States of America
* Emily.Ho@oregonstate.edu

**Data Availability Statement:** All raw sequence read files are available from the NCBI SRA database (bioproject accession number PRJNA831825). Generalized linear modeling results are available in

## Abstract

Older adult populations are at risk for zinc deficiency, which may predispose them to immune dysfunction and age-related chronic inflammation that drives myriad diseases and disorders. Recent work also implicates the gut microbiome in the onset and severity of age-related inflammation, indicating that dietary zinc status and the gut microbiome may interact to impact age-related host immunity. We hypothesize that age-related alterations in the gut microbiome contribute to the demonstrated zinc deficits in host zinc levels and increased inflammation. We tested this hypothesis with a multifactor two-part study design in a C57BL/6 mouse model. The two studies included young (2 month old) and aged (24 month old) mice fed either (1) a zinc adequate or zinc supplemented diet, or (2) a zinc adequate or marginal zinc deficient diet, respectively. Overall microbiome composition did not significantly change with zinc status; beta diversity was driven almost exclusively by age effects. Microbiome differences due to age are evident at all taxonomic levels, with more than half of all taxonomic units significantly different. Furthermore, we found 150 out of 186 genera were significantly different between the two age groups, with *Bacteriodes* and *Parabacteroides* being the primary taxa of young and old mice, respectively. These data suggest that modulating individual micronutrient concentrations does not lead to comprehensive microbiome shifts, but rather affects specific components of the gut microbiome. However, a phylogenetic agglomeration technique (ClaaTU) revealed phylogenetic clades that respond to modulation of dietary zinc status and inflammation state in an age-dependent manner. Collectively, these results suggest that a complex interplay exists between host age, gut microbiome composition, and dietary zinc status.

Supporting Information S3, S4, and S5 Tables. Code to generate the results is available at https://github.com/davised/davis-2022-ZAM.

**Funding:** Funding for this work was provided by the United States Department of Agriculture (USDA) - National Institute of Food and Agriculture (NIFA) and the National Sciences Foundation (NSF). E.H. was funded by NIFA grant 2018-67017-27358 and AES-W4002. https://www.nifa.usda.gov/. T.J.S. was funded by NSF grant #1557192. https://nsf.gov/. The funders had no role in study design, data collection and analysis, decision to publish, or preparation of the manuscript.

**Competing interests:** The authors have declared that no competing interests exist.

## Introduction

A hallmark of aging is a progressive increase in chronic inflammation. As this chronic inflammation precedes the onset of myriad age-related disorders, researchers have sought to uncover the factors that drive its development. Two such factors that have emerged in recent years and which are known to interact and impact one another are micronutrient deficiency—especially dietary zinc—and the gut microbiome. It is important to understand the factors that structure the gut microbiome, as doing so clarifies the etiology of microbiome-mediated diseases, aids in the appropriate contextualization of microbiome diagnostics, and improves consideration of factors or processes that may confound the efficacy of microbiome therapeutics.

Extensive research clarifies how different exogenous and endogenous factors can shape the microbiome. Aging has been shown to be an especially relevant endogenous dictate of the temporal dynamics of the gut microbiome. Human infants display high intrapersonal variability in their gut microbiome, with large scale diversification occurring between birth and 3 years of age [1, 2]. Furthermore, bacterial diversity increases with age in individuals regardless of other studied covariates [1]. In addition to its rapid and substantial diversification early in life, the gut microbiome also experiences diversification, albeit more of a gradual form, amongst aged individuals (e.g., over 65 years of age) [3]. With respect to exogenous factors, diet has been shown to be one of the key determinants of microbiome composition [4, 5]. The gut microbiome has been shown to respond relatively quickly and strikingly to changes in diet, though most of this work has focused on comparing the effects of fundamentally different compositions of macronutrients on the gut microbiome [6, 7]. The limited work in response to micronutrients, such as vitamins and minerals, tends to indicate that changes in micronutrient composition can impact the gut microbiome, but with a far less substantial effect [8].

Research that has defined the impact of specific endogeneous and exogeneous factors on the gut microbiome have largely done so without considering the potential interaction between factors. Recent research demonstrates the importance of considering interactions: combining factors can elicit alterations to the composition of the microbiome that do not occur in isolation. For example, reducing dietary zinc sensitizes the mouse gut microbiome to oral arsenic exposure [9]. Additionally, the use of genetically manipulated mice revealed that the aryl hydrocarbon receptor (AHR) interacts with dietary macronutrient content to impact gut microbiome composition in unique ways [10]. Furthermore, dietary fat content impacts the mouse gut microbiome's resistance to antibiotic exposure [11]. During pregnancy, zinc deficiency may have additional effects on the gut microbiome [12]. These and other studies point to the importance of considering how the combination of factors impact the microbiome, and doubly so when those factors are known to interact in a health-related context, such as the case of age-related inflammation and micronutrient deficiency.

Extensive research has explored the link between aging, diet, and age-related phenotypes like inflammation, but how age and diet interact to influence microbiome composition is unknown. The value of proper nutrition to healthy aging is well established [13, 14]. Inadequate nutrition is a common risk factor to age-related diseases and the general decline in health with age [15, 16]. Aged individuals are often more susceptible to micronutrient deficiencies, particularly zinc, due to both decreased food intake, and reduced absorption of such micronutrients, which compounds risk for micronutrient issues [17, 18].

In an effort to understand the susceptibility to micronutrient deficiencies, we hypothesize that the microbiome is a potential source of age-related micronutrient variation, given that aging has a measurable impact on the microbiome. For example, as microbes in the gut have a metabolic requirement for zinc, the gut microbiome may contribute to changes in zinc availability as their hosts age. Additionally, modulating dietary zinc status has been shown to result

in significant changes to the host gut microbiome [19]. As aged individuals have an increased requirement for zinc and have a gut microbiome that is distinct from young adults, studying the interaction between age, zinc status, markers of inflammation, and the microbiome is a promising avenue for understanding the role of zinc in healthy aging.

To disentangle the link between the gut microbiome, aging and inflammation, and zinc status, we conducted two mouse model experiments. Both experiments compared young and aged mice, but varied in the amount of dietary zinc the mice were exposed to. Specifically, in the first experiment, we compared individuals fed either a zinc adequate (ZA) or zinc supplemented (ZS) diet to determine how the effect of increasing dietary zinc impacts the gut microbiome and whether it does so differently as a function of age. In the second experiment, we compared individuals fed either a ZA or marginal zinc deficient (ZD) diet, which allowed us to evaluate how reducing dietary zinc impacts the gut microbiome and whether it does so differently as a function of age. Host metabolic and immune marker data were examined and described in a recently published work, where we found significant increases in lipopolysaccharide-induced interleukin-6 (IL-6) in ZD old mice compared to ZA old mice controls, and significant reductions in several markers of inflammation, including plasma monocyte chemoattractant protein-1 levels, T cell activation-induced Interferon-gamma, interleukin-17, and tumor necrosis factor alpha response, and increased naive CD4+ T cell subset in ZS old mice compared to the ZA old mice controls [20]. Here, we describe the microbiome results of these experiments, as well as their integration, to clarify the impact of aging, inflammation state (using IL-6 measurements), and dietary zinc state on the gut microbiome.

## Methods

### Animal husbandry and sample collection

Young C57Bl/6 male mice were purchased from Jackson Laboratories (Bar Harbor, ME). Old C57Bl/6 male mice were obtained from the National Institute of Aging (NIA) aged rodent colonies. To mitigate potential facility effects, mice were purchased from different NIA aged colony locations (Kingston Division, Raleigh Division, Frederick Division), and from different rooms (Jackson Laboratories mouse colonies). To control for the potential effects of switching from a standard chow diet (irradiated PicoLab LabDiet 5053) to a purified AIN-93 rodent diet, all mice were acclimated to the control zinc adequate diet (see diet details below) for 1 mo prior to the beginning of the study. After acclimation, young mice (2 mo) and aged mice (24 mo) were randomly assigned by animal handler to different dietary treatments for 6 wks. All mice were individually housed in ventilated microisolater cages and kept in a temperature and humidity controlled environment (72°F, 50% humidity, 12-h light cycle). Upon researcher examination, all aged mice had no obvious signs of illness, tumors, or lesions at the beginning of the study.

To study the effects of marginal zinc deficiency, groups of 10 young and old mice were fed a purified diet containing either 30 mg/kg zinc (zinc adequate, ZA), or 6 mg/kg zinc (marginal zinc deficient, ZD) for 6 wks. To study the effects of zinc supplementation, groups of 10 young and old mice were fed a purified diet containing either 30 mg/kg zinc (ZA), or 300 mg/kg zinc (zinc supplemented, ZS) for 6 wks. A total of 80 mice were used in this study (40 in marginal zinc deficiency study, 40 in zinc supplementation study). The number of mice in each group was chosen to result in 99% power to detect differences in gut microbiome community composition or animal mineral status between treatment groups (two-way ANOVA; alpha = 0.05). Additionally, this group size results in 90% power to detect differences in the relative abundance of phylotypes in the microbiome that differ by at least half an order of magnitude in relative abundance between groups. Data from all mice were used (no exclusion) in data analyses. Additional clarification of mouse handling, study design, and conclusions based on the host data can be found in S1

Checklist. Adequate zinc levels aligned with recommended zinc levels in AIN-93G rodent diets and marginal zinc deficiency and zinc supplementation models have been established previously [9, 21]. Diets were formulated using a modified egg white-based AIN-93G diet wherein zinc was provided as zinc carbonate as previously described. Purified ZA, ZD, and ZS diets were purchased from Research Diets (New Brunswick, NJ). The ZD diet and ZS diet had previously been shown to result in reduced zinc status and increased zinc status in mice, respectively [21, 22]. Food and water were provided *ad libitum*, and food intakes and body weights of all mice were monitored throughout the study. Fecal samples were collected every two weeks throughout the study duration, and were stored at -80C. At the beginning (week 0) and end (week 6) of study, whole blood samples were collected from the submandibular vein in all mice using Goldenrod animal lancets (Braintree Scientific, Braintree, MA) using BD Microtainer heparin blood collection tubes (BD Biosciences, Franklin Lakes, NJ). Measurement of whole blood IL-6 has been previously described [20]. Briefly, heparinized whole blood samples were stimulated with 10 ng/ml LPS for 24 h. Production of IL-6 was measured using BD Cytometric Bead Array Mouse Inflammation Kit (BD Biosciences). At the termination of the experiments, mice were euthanized by $CO_2$ asphyxiation, and sera and tissues were collected. All mice were monitored regularly for potential adverse effects. Mice that exhibited signs of illness as defined in the approved animal protocol were humanely euthanized and removed from the study upon consultation with the attending veterinarian. The animal use protocol was approved by the Oregon State University (OSU) Institutional Laboratory Animal Care and Use Committee, and adheres to the international standards of animal care as established by the Association for Assessment and Accreditation of Laboratory Animal Care International. Host physiological data, including serum zinc and LPS-induced IL-6 (S6 Table), were measured as explained in Wong et al. 2021 [20].

## Microbiome data generation

Fecal DNA was extracted using QIAamp PowerFecal DNA Kit according to the manufacturer's protocol (Qiagen, Germantown, MD). Fecal samples at week 0 and week 6, and the cecum and colon samples collected at week 6, post humane euthanasia, were used for targeted 16S DNA sequencing; fecal sample data was analyzed in the current study and cecum and colon data were deposited into the NCBI database for recording purposes. Following our prior work, the rDNA 16S V4 region was targeted for amplification and sequencing, following the Earth Microbiome Project (EMP) library preparation and sequencing protocol; the 515F-806R primer pairs were used [9, 23]. The 16S libraries were sequenced at the Center Quantitative Life Sciences at OSU on the Illumina MiSeq platform with the 2x300 sequencing reaction, with mean sequencing depth of 130,000 reads per sample (Illumina, Inc. San Diego, CA). The EMP 16S Read 1, Read 2, and Indexing primers were used during the sequencing reaction, leading to exclusion of the 16S primer sequence in the raw sequencing reads.

## Amplicon sequence variant (ASV) calling

Initial sequence quality was checked using FastQC (v0.11.3) [24] (https://www.bioinformatics. babraham.ac.uk/projects/fastqc/). Forward read and reverse read average read qualities dropped below Q30 at approximately position 240 and 200, respectively. Forward and reverse reads were processed using the DADA2 (v1.8) pipeline from Bioconductor (v3.7) in R (v3.5) [25, 26]. Non-default settings for the filterAndTrim() function were truncLen = c(240, 200), maxN = 0, maxEE = c(2, 2), rm.phix = TRUE. As the EMP protocol sequencing primers were used, no PCR amplicon primers were identified at the 5' ends of the R1 and R2 reads; truncation during the filterAndTrim() step removed any primers and/or adapters present at the 3' ends. Sequences were restricted to a final length of 250-256bp after merging.

Taxonomy and species assignment for each ASV was done using the Silva (v132) training and species identification datasets with the naive bayesian classifier built-in to dada2 [27–29]. Mitochondrial and chloroplast sequences, if present, were removed. A prevalence threshold of five samples (half an experimental unit [n = 10]) was used to filter low prevalence ASVs, which yielded 361 ASVs remaining in the analysis. Unknown taxonomic assignments at the genus level (i.e. 'NA') were assigned placeholder taxonomic names with the form of FamilyName_-Genus_ASV# such that we balance overgrouping (i.e. grouping all ASVs of a particular family designation at the genus level) and excluding data (i.e. discarding any ASVs that have no genus designation). The practical effect of this is that each ASV with an unknown genus is considered its own genus when doing genus level comparisons.

## Microbiome data analysis

ASV and taxonomy tables were imported into a phyloseq object for filtering and analysis [30]. UpSetR v1.4.0 was used to generate the upset plots showing shared ASVs by group [31]. Stacked bar plots were generated using phyloseq and modified using ggplot2 [30, 32].

Robust principal component analysis (rpca) from DEICODE was done in standalone mode with default parameters [33]. Permutational multivariate analysis of variance (PERMANOVA) comparing the centroids and permutation tests comparing the group dispersions were done using the *adonis()* and *permutest.betadisper()* functions of vegan v2.5–7, respectively [34]. Non-parametric tests (i.e. Wilcoxon Rank Sum tests and Kruskal-Wallis tests), were done in R v4.1.2 using the wrapper functions *wilcox_test()* and *kruskal_test()* from the rstatix v0.7.0 package [26, 35]. All tests were corrected for multiple comparisons using a false discovery rate (fdr) p-value cutoff of 0.05 [36].

## Phylogenetic tree construction and cladal analysis

As in our prior work [37–39], we assembled a phylogenetic tree from rarefied ASV sequences. ASV sequences were aligned, alongside 100 phylogenetically diverse full length 16S rDNA gene sequences (i.e., guide sequences) obtained from the SILVA all species living tree project v1.2.3, to the SILVA reference alignment using the mothur v1.39.3 implementation of the NAST aligner (*flip = T*). Previous work indicates that inclusion of full-length phylogenetically diverse guide sequences improves phylogenetic accuracy of reconstruction [37, 40]. Phylogenetic tree construction was performed using FastTree v2.1.10 (*-nt -gtr*) [41]. We used a custom R script v4.0.3 to prune guide sequences from the resultant phylogenetic tree in order to provide a custom phylogeny with only ASVs in our study of interest *(drop.tip())* using the ape R package [42]. The resultant tree was midpoint rooted (*midpoint.root()*) using the phytools v0.7.70 R package [43]. We used the Cladal Taxonomic Unit (ClaaTU) algorithm to determine clades of microbes which manifested significant statistical relationships with diet and physiologic covariates [37]. In brief, the ClaaTU algorithm performs a root-to-tip tree traversal of the phylogenetic tree. For each monophyletic clade within the tree, all descendent tips are summed to result into a matrix *ctu* where an entry *ctu[i,j]* corresponds to the abundance of clade *j* within sample *i*.

We ran linear regressions to identify significantly differential abundances, one for age (a generalized linear model under the negative binomial distribution) and one for diet (a generalized linear mixed model, also under the negative binomial distribution). The rarefied abundance counts for each node in the fecal week 0 and week 6 samples were tested using the *glm.nb(formula('Abundance ~ Age'))* for age (from MASS) and the *glmer.nb(formula('Abundance ~ Diet + (1 | Host)')* for diet (from lme4) function in R [26, 44]. For the diet models, to control for age and study effects, one model was run for each age by diet by study group (four total models). Additionally, we ran two generalized linear models (negative binomial distribution as

in the MASS package [26]) to identify differential abundances correlating with inflammation marker IL-6 in the week 6 samples using *glm.nb(formula('Abundance ~ wk6_WB_LPS_IL6'))*. To control for differences in response to inflammation between age groups, and to look at effects at the end of the study, we only examined the week 6 abundances, and generated one model for each age group. Significant coefficients in all regressions were identified using the Wald test. Wald test p-values were corrected for multiple testing by using the Benjamini & Hochberg false discovery rate correction at a 0.05 significance level [36]. Significant nodes were excluded if the predicted model had a negative value for the Intercept as count-based methods have zero as a lower bound.

## Results

### Age is the most important factor distinguishing gut microbiome content of mice

To examine the effect of zinc supplementation and deficiency on gut microbiome content, we fed C57Bl/6 (young = 2mo, old = 24mo) mice food containing differing zinc concentrations (deficient [ZD] = 6mg/kg, adequate [ZA] = 30mg/kg, supplemented [ZS] = 300mg/kg zinc). Due to limits on acquiring aged mice, the supplementation and deficiency studies were conducted separately, with ZA mice present as controls in both studies (supplementation and deficiency studies will be referred to as **z**inc and **a**ging **m**ouse 1 and 2 [ZAM1 and ZAM2, respectively]). Fecal samples were taken at week 0 and week 6 of the study and the microbiome content was determined using 16S amplicon sequencing.

Analysis of beta diversity using robust Aitchison principal component analysis (rpca) indicated that the majority of the variance in the gut microbiome can be explained by mouse age. This result was replicated in both studies and when the two studies were combined (Fig 1, panels A-C). PERMANOVA confirmed statistical significance between centroids (R2 = 0.918, p = 0.001); a permutation test of the beta dispersions showed evidence for differences in variance between the two age groups in the combined data set, while there was no difference found in interstudy variance when examining all samples simultaneously (p = 0.002 and p > 0.05, respectively). However, a more refined beta dispersion analysis focusing on each age group within each study individually did indicate that there are differences in dispersion, with the ZAM2 samples having less dispersion than those from ZAM1 (S1 Fig, p = 0.001, p = 0.018). Despite the significant differences in dispersion, we opted to analyze the combined ZAM1 and ZAM2 datasets together for the remainder of the analysis, while implementing controls in testing that allowed for us to account for the inter-study differences.

These beta-diversity results indicated that specific taxa stratify young and aged mice, possibly even in ways that are robust to dietary variation. These differences are visualized in the stacked bar chart with ASVs agglomerated at the genus level, and grouped by study and diet (Fig 1D). At the genus level, we found a stark difference between the young and old mice, with *Bacteroides* and *Parabacteroides* as the primary genera in young and old mice, respectively, making up around 50% of the gut microbiome content on average. A Wilcoxon Rank Sum test between old and young mice confirmed the difference (adjusted *p* < 0.05) in both *Bacteroides* and *Parabacteroides*. Furthermore, of the 186 total genus level groups in the dataset, including the *Bacteroides* and *Parabacteroides*, 150 were significantly different between the age groups (S1 Table). The top 12 most abundant genera, limited to those averaging > 0.5% relative abundance and totaling 75–100% of abundance per sample, are shown in Fig 2. These data suggest that while the most abundant components of the gut microbiome are dependent on age, many of the less abundant taxa vary in response to age as well. Moreover, evaluation of these age effects across the taxonomic hierarchy indicates that these effects are not relegated to a specific taxonomic rank or unit (S2 Table).

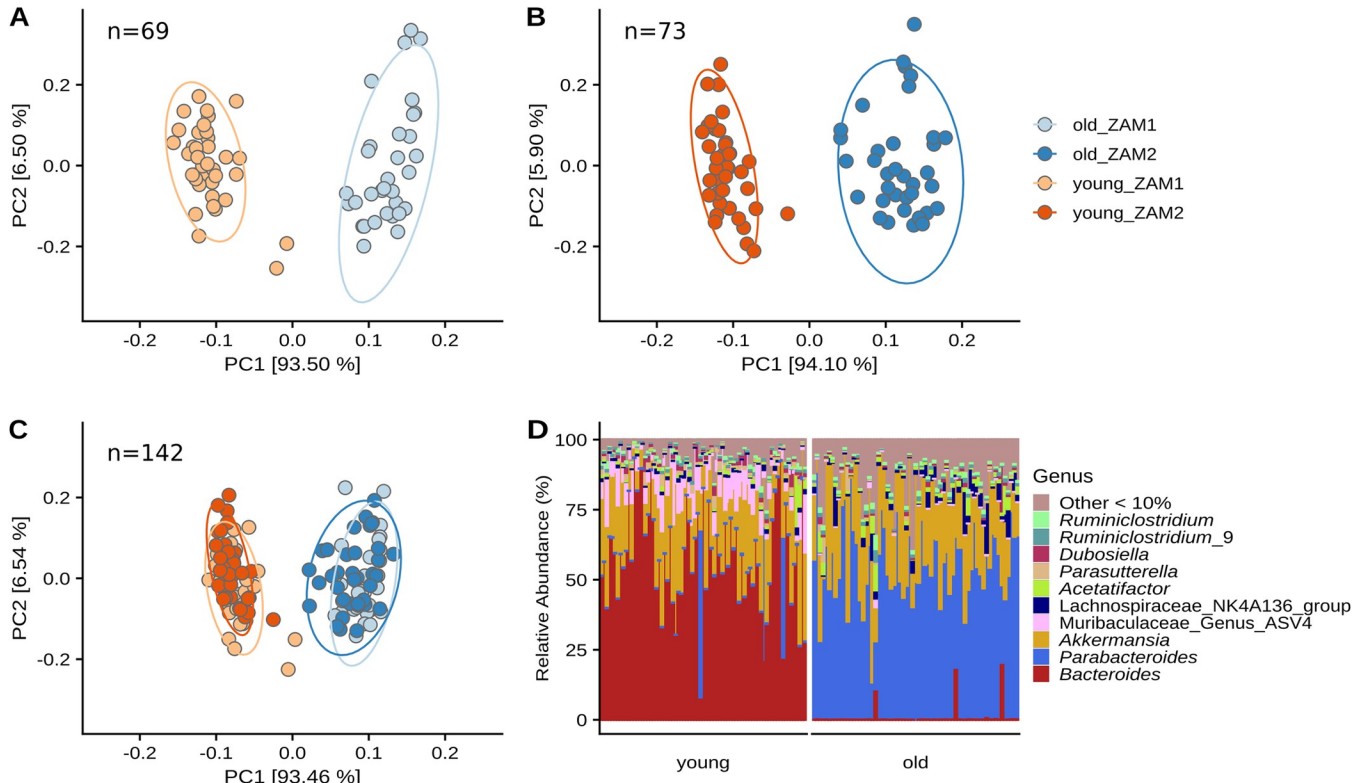

**Fig 1. The majority of the variance in the mouse gut microbiome can be explained by age.** Robust principal component analysis results are shown for ZAM1 (zinc supplemented), ZAM2 (zinc deficient) and combined, respectively (A-C). Age explains the majority of the variance in beta diversity in both studies. Relative abundances of ASVs agglomerated at the genus level are shown (D). *Bacteroides* and *Parabacteroides* are the most abundant genera in young and old mice, respectively. n = 69 for ZAM1, n = 73 for ZAM2, and n = 142 for combined beta diversity and stacked barplot analysis of week 0 and week 6 fecal samples.

Given the strong effect of age on the composition of the gut microbiome and absence of an age-robust effect of dietary variation, we next asked if there exist taxa that appear to be sensitive to dietary variation in an age-dependent manner. We used a Kruskal-Wallis test to determine if specific taxa differed in their relative abundance across diet groups while blocking on age. Further, to examine the differences that may have occurred over the study period, we restricted the comparisons to the week 6 fecal samples only. Applying this test to each genus resolved no significant differences in either age group (all fdr adjusted p-value > 0.05). Collectively, these results indicate that specific phylotypes are not evidently sensitive to dietary zinc status regardless of analytical controls on age.

Having exhausted the non-parametric tests for significant differences within and between age and diet groups, we opted to use a phylogenetic agglomeration method to further examine the data. In this way, we may avoid any biases imposed by using a predefined taxonomic grouping and instead use the phylogenetic groups found in our own dataset.

## Phylogenetic aggregation reveals clades that vary with age and dietary zinc status

We next considered that the effects of age and dietary zinc status on gut microbes may be subject to phylogenetic redundancy, in such that close evolutionary relatives manifest shared traits that are sensitive to these changes in the host or diet. Because such taxa may displace one

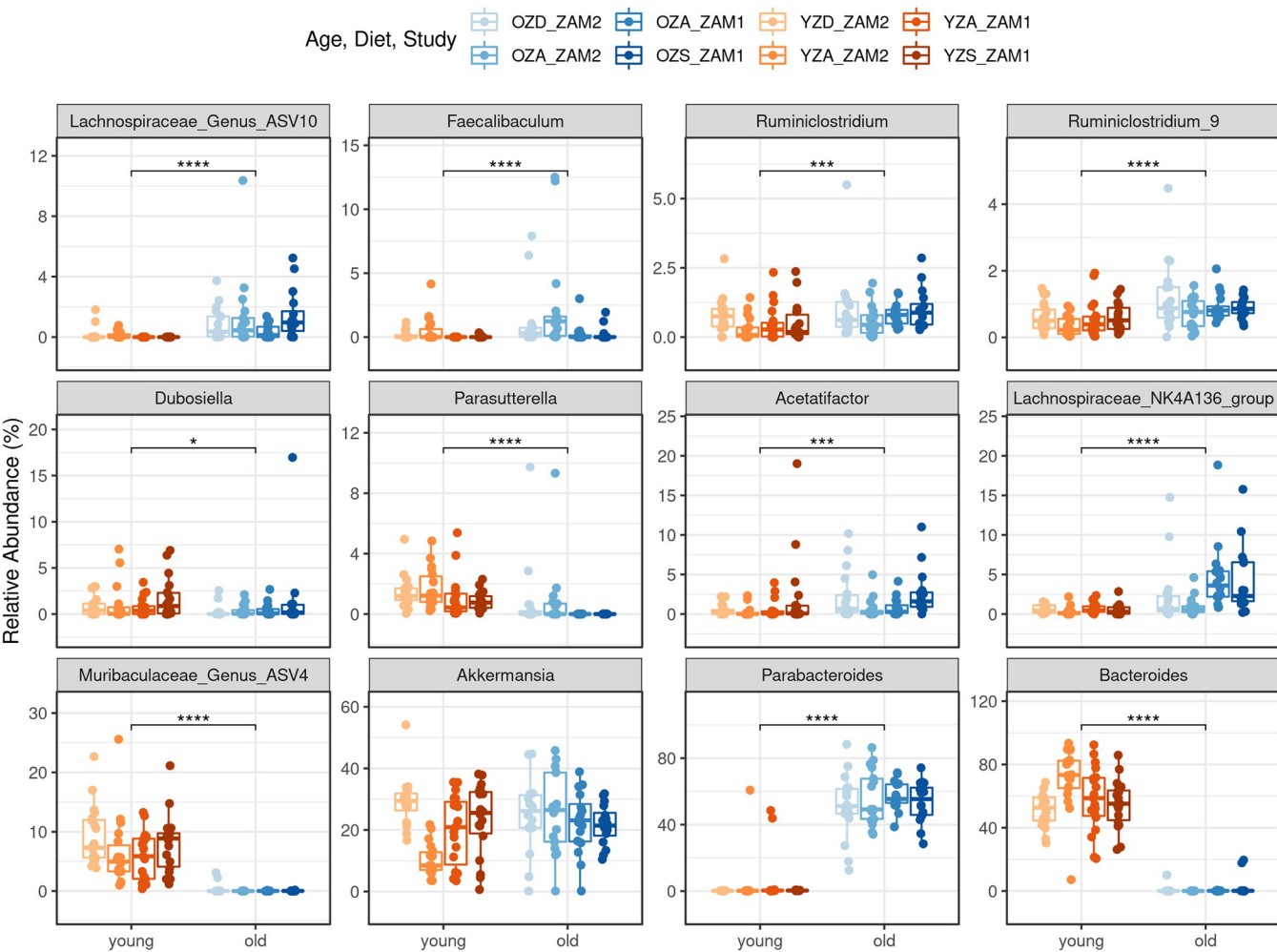

**Fig 2. Eleven of the top 12 most abundant genera are significantly different between young and old mice.** Boxplots display relative abundance of each of the top twelve most abundant genera in the study, with the first, median, and third quartiles shown by the hinges, and the whiskers extend to no more than 1.5 x inter-quartile range, with outliers plotted individually. Additionally, each data point is shown as a jittered point value. Boxplots and points are colored based on age by diet by study groups, as shown. Significance values are shown based on a Wilcoxon Rank Sum test between young and old mice, with *, **, ***, and **** corresponding to p-values of 0.05, 0.01, 0.001, and 0.0001, respectively. ZAM1—supplementation study, ZAM2—deficiency study, OZD—old zinc deficient, OZA—old zinc adequate, OZS—old zinc supplemented, YZD—young zinc deficient, YZA—young zinc adequate, YZS—young zinc supplemented. n = 75 young mice fecal samples, n = 67 old mice fecal samples, including week 0 and week 6 fecal samples.

another in the gut, traditional phylotyping analysis may obscure our ability to resolve the association between these related groups of taxa (i.e., clades) and either age or dietary zinc status. To explore this hypothesis, we used a phylogenetic agglomeration method, called Cladal Taxonomic Unit (ClaaTU) analysis, which groups ASVs into monophyletic clades and subsequently evaluates how the abundance of the clade associates with host covariates [37]. In particular, we assembled a phylogenetic tree relating ASVs, determined the empirically derived phylogenetic clades using ClaaTU, and calculated the rarefied abundances per clade as input for a generalized linear model (glm) under a negative binomial distribution.

We first evaluated how clades link to the two age groups. When regressing clade abundance as a function of age, we find that of the 360 clades in the phylogeny, 240 of them have a significant age term (S3 Table; Wald test; fdr adjusted p-value < 0.05). This observation is consistent with the results when examining phylotypes as reported above (S1 Table); the clades that link

to age are nested within higher order phylotypes that our Wilcoxon Rank Sum analysis detected as being linked to age. Of the ten clades (count in parentheses) with the largest magnitudes of beta coefficients, those associated positively with young mice are all of the family Muribaculaceae (3). Clades negatively correlated with young mice are in family Lachnospiraceae (2), genus Lachnospiraceae UCG-006 (1), family Ruminococcaceae (1), and genus *Acetatifactor* (2). These observations indicate that a diverse collection of bacteria have traits that are sensitive to host age and that the traits in question evolved at multiple and distinct points across the bacterial phylogeny.

We next sought to uncover phylogenetic clades that link to dietary zinc status. In order to disambiguate any effects of study, we took an approach that applied four individual generalized linear mixed models (glmm), wherein each model represented a specific age group within a specific diet trial, that all followed the same form. Our model also leveraged the longitudinal design of our study (week 0 and week 6 observations) by including a random effect that controlled for repeated measures in the same mouse. This model design allowed us to compare the zinc-adjusted diets to the control diet for each of the age groups. These models also ensured that the discoveries they produce result from changes in relative abundance as a result of dietary intervention, rather than differences in pre-intervention relative abundance between groups. We identified 39 significant clades through this approach, which are displayed on the phylogenetic tree (Fig 3; S4 Table).

Our analysis points to 15 of the 39 significant clades varying as a result of zinc supplementation. We found that zinc supplementation is correlated with changes in two different clades found within the Lachnospiraceae, which is the richest family in the phylogeny (Lachnospiraceae contains 169 of the 361 ASVs detected across both studies). These two clades manifest opposing associations with zinc supplementation and age dependency. One clade, made up of

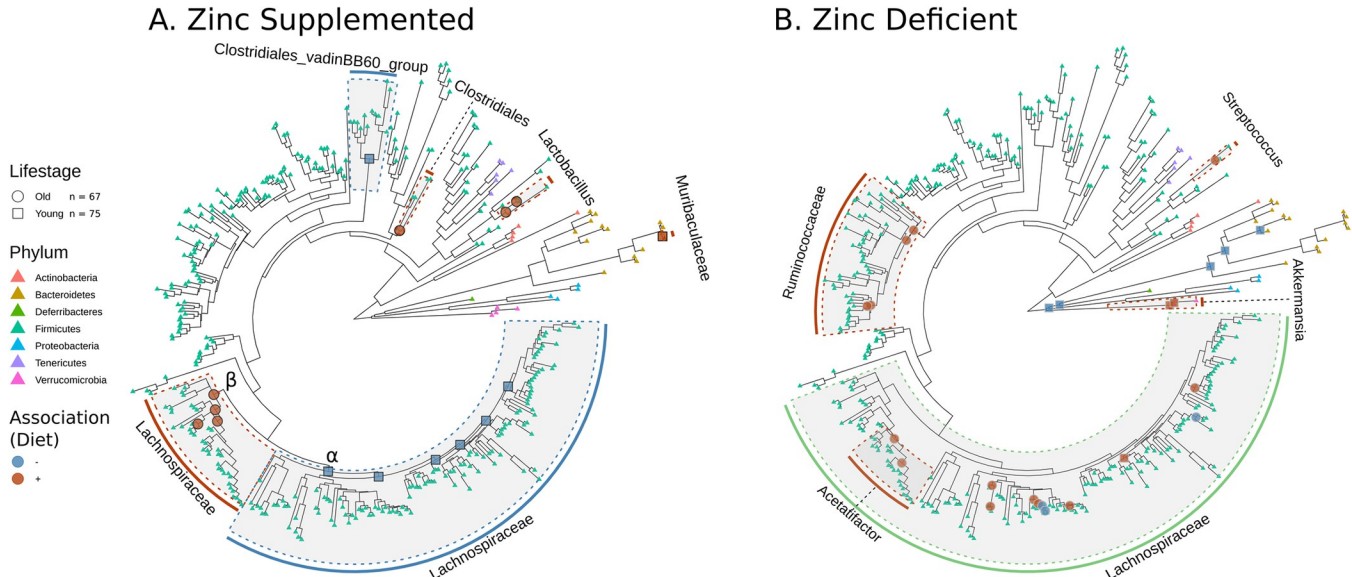

**Fig 3. Associations between bacterial cladal abundance and dietary zinc supplementation and deficiency.** The phylogenetic tree (identical for both A and B) is based on alignment of ASVs to a reference 16S alignment. Clades are marked with host lifestage (circles for old and squares for young mice, respectively) and type of association (blue for negative, orange for positive) with dietary zinc supplementation (A) and deficiency (B). Shaded clades are marked with the respective shared taxonomic phylotype, and color coded based on association (green indicates variable association within a particular clade). Tree tips are colored based on taxonomic phylum, regardless of dietary zinc status. α and β indicate Lachnospiraceae clades that associate with zinc supplementation in a negative and positive way, respectively. n = 75 young mice fecal samples (37 in supplementation study, 38 in deficiency study), n = 67 old mice fecal samples (32 in supplementation study, 35 in deficiency study), using both week 0 and week 6 fecal samples.

four total nested subclades, shows a positive correlation in old mice (indicated by β in Fig 3); the other clade, made up of six total nested subclades, shows a negative correlation in young mice (indicated by α in Fig 3). Moreover, we find that the young mice also experience a significant increase in a single clade with members of the Muribaculaceae and a decrease in a clade with members of the Clostridiales vadinBB60 group when fed zinc supplemented diets. Additionally, the aged mice fed zinc supplemented diets have significant increases in two nested clades within *Lactobacillus* and a clade including members of the order Clostridiales.

Zinc deficiency resulted in more varied responses among 24 significant microbial clades. Eleven clades within the Lachnospiraceae have a significant association with zinc deficiency. Interestingly, a set of two nested clades have a positive association with zinc deficiency in aged mice, yet two additionally nested clades show a negative association with zinc deficiency. This effect potentially indicates a limited space in the microbiome for a particular functional niche whereby an increase in one set of members of the Lachnospiraceae occurs with a decrease in another clade that occupies that same functional niche. Two clades of the *Acetitafactor* genus, itself a subclade of the Lachnospiraceae, have a positive correlation with zinc deficiency in the aged mice. Four nested clades within Ruminococcaceae and a single clade within the *Streptococcus* genus also positively correlate with zinc deficiency in the aged mice. Two clades of *Akkermansia* and five clades that include multiple genera (*Bacteroides* as the most notable) have a positive and negative association, respectively, with zinc deficiency in the young mice.

## Clades of microbiota associate with a marker of inflammation

Chronic inflammation associated with age, coined "inflammaging", is one of the hallmarks of aging [45]. Increased inflammatory cytokines, including increased IL-6, has been shown to be associated with age [46, 47]. In our previous work, LPS-induced IL-6 response in whole blood increased with zinc deficiency, and was further increased with age [20]. Therefore, in order to examine the effect of inflammation on the gut microbiome in young and old mice, we also sought to identify gut microbes that link to IL-6. To do so, we generated a generalized linear model for each clade, looking at only the week 6 abundances in young and old mice across dietary groups (Fig 4; S5 Table). We also identified clades that had an intersection between the models for inflammation and the models for zinc dietary effects (Squares and Triangles, Fig 4). We found many clades within Lachnospiraceae are correlated with IL-6 in both young and old mice. Interestingly, the same clade that is negatively correlated with zinc supplementation in young mice is positively correlated with IL-6 in young mice (see α in Figs 3 and 4). This evidence supports the hypothesis that zinc supplementation can potentially mitigate the effects of inflammation, albeit in young mice. Thirty-three clades of Lachnospiraceae, including ten *Acetatifactor*, are positively associated with IL-6. Conversely, twenty-two Lachnospiraceae clades, including five *Acetatifactor* clades, are negatively associated with IL-6. Moreover, three clades of Muribaculaceae, including two shared between both young and old mice, are negatively associated with IL-6; one clade is positively associated with IL-6 in old mice only. Muribaculaceae are found in higher abundances in young mice (Fig 2; S3 Table), so these results indicate that Muribaculaceae may putatively mediate inflammation within young mice. Additionally, a clade of *Faecalibaculum* is negatively correlated with IL-6 in both old and young mice (see β in Fig 4). Finally, three nested clades within the Clostridiales are positively associated with IL-6 in young mice, while three other nested clades within the Clostridiales, specifically members of *Tyzzerella* and ASF356, are negatively associated with IL-6 in old mice (see γ in Fig 4).

We also observed clades that associated with IL-6 in either young or old mice alone. A clade within the most dominant taxa in the young mice, *Bacteroides*, is negatively correlated with IL-

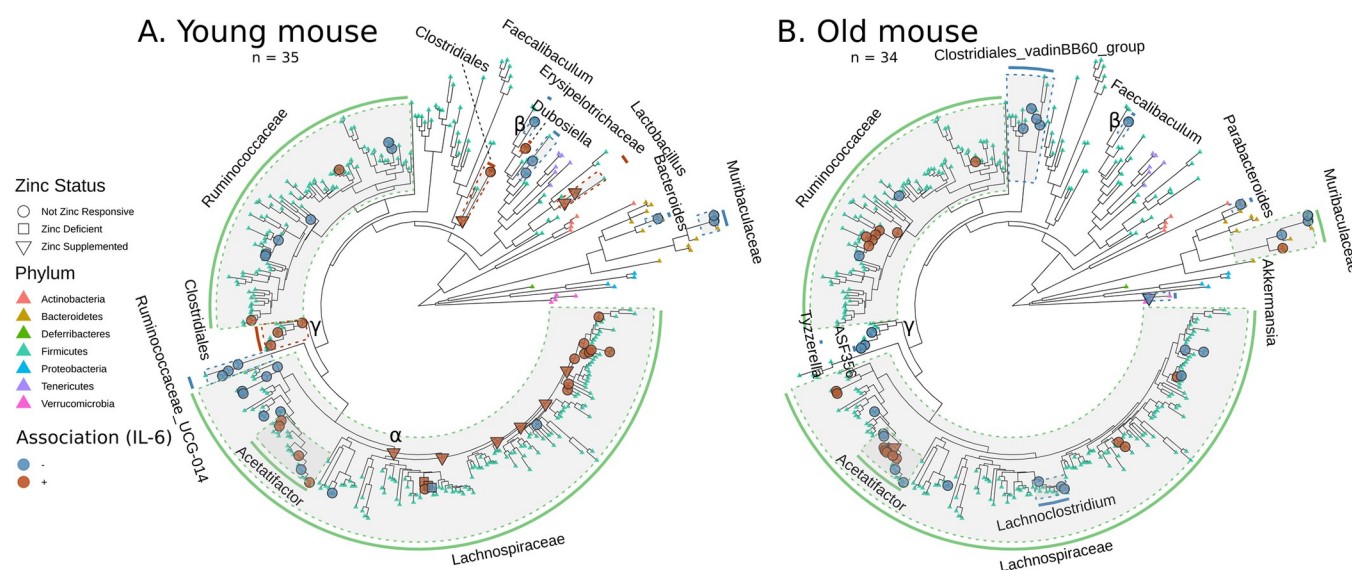

**Fig 4. Associations between bacterial cladal abundance and inflammation marker IL-6.** The phylogenetic tree (identical for both A and B) is based on alignment of ASVs to a reference 16S alignment. Clades are colored for type of association (blue for negative, orange for positive) with IL-6 for young (A) and old (B) mice, respectively. Nodes are marked with squares, triangles, and circles for overlap with significant associations with zinc deficiency, zinc supplementation, and no overlap, respectively. Shaded clades are marked with the respective shared taxonomic phylotype, and color coded based on association (green indicates variable association within a particular clade). Tree tips are colored based on taxonomic phylum, regardless of association with IL-6. α indicates the most basal clade of *Lachnospiraceae* that is negatively correlated with zinc supplementation and positively correlated with IL-6. β indicates a clade of *Faecalibaculum* that is negatively correlated with IL-6 in both young and old mice. γ indicates the basal node of three clades each that are positively and negatively correlated with young and old mice, respectively. n = 35 young mice fecal samples, n = 34 old mice fecal samples, using only week 6 fecal samples including from both supplementation and deficiency studies.

6, which may indicate that loss of *Bacteroides* in the young mouse gut is indicative of a mouse suffering from inflammation. In young mice, six clades of Ruminococcaceae and three clades of Ruminococcaceae UCG-014 are negatively correlated with IL-6, with only two Ruminococcaceae being positively correlated with IL-6. Conversely, seven clades of Ruminococcaceae are positively correlated with IL-6 in old mice, with only two clades showing a negative correlation. These differences within a broader phylogenetic clade indicate that specific subclades within a particular higher-order group may have age-specific effects in determining host inflammatory response. In addition to the clade of Clostridiales mentioned above, two additional clades of Clostridiales, one from *Dubosiella*, and two from *Lactobacillus*, are positively correlated with IL-6 in young mice. Interestingly, the two clades of *Lactobacillus* are also positively correlated with zinc supplementation in old mice, not young mice, indicating a complex interplay between host inflammation and zinc status. In old mice, four clades of Clostridiales vadinBB60 group and a single clade of *Akkermansia* are negatively correlated with IL-6, while no correlations with those clades were found in young mice. Collectively, these results underscore the interaction between aging, inflammation, and the gut microbiome.

## Discussion/Conclusions

### Age effects shape the overall structure of the mouse gut microbiome

Many investigations that have surveyed the effect of diet on the microbiome have found that diet is a major determinant of microbiome composition. But, these studies typically compare diets that are fundamentally different in the macronutrient composition [5, 6, 48]. Here, we evaluated how modulating a single dietary micronutrient—zinc—impacts the composition of the gut microbiome in an aging mouse model. Our research focused on zinc in part because of

its role in immune health and the fact that zinc deficiency is relatively common among aged individuals [14, 15]. Additionally, in a different model system (broiler chicken; *Gallus gallus*), drastic changes in microbial composition were seen in zinc deficient hosts [19]. These changes in *Gallus gallus* are similar to those seen in chickens with different physiological diseases [19]. Moreover, our prior work indicates that dietary zinc status sensitizes the gut microbiome to secondary perturbations in ways that link to physiology, but does not necessarily correlate with drastic rearrangements of the host microbiome [9]. We sought to build upon this background to ascertain if modulating dietary zinc status could normalize the effects of aging on the gut microbiome or the microbiome-inflammaging axis.

We find that aging elicits a substantial effect on the composition of the mouse gut microbiome, more so than nutritional zinc status or study effects. Reasons for this variation could include age-dependent changes in physiology that select for fundamentally different microbial communities or that micronutrient variation may in general only have very modest impacts on microbiome composition [8, 49]. Indeed, prior work has underscored the variation in the composition of the microbiome across human lifespan and between young and aged mice [3, 50, 51]. Prior work also points to intestinal inflammation and gut barrier integrity as being an age-dependent factor that drives the successional dynamics of the microbiome late in lifespan [52, 53].

We show here that these age-specific differences may contribute to age-specific effects in microbiome content independent of the immune modulating dietary micronutrient zinc, at least over the range tested in our studies, and that age-related differences in the microbiome are repeatable across multiple experiments. These observations suggest that mouse aging serves as an important driver of the gut microbiome and that investigations on the impacts of the diet on the gut microbiome may need to consider the age-dependent context of their observations.

## Age and dietary zinc status interact to affect specific gut microbiota

While age effects dominate the overall composition of the microbiome and variation at the genus to phylum levels, phylogenetic aggregation revealed monophyletic clades of taxa that vary as a function of dietary zinc status. However, the effect of zinc status on these taxa differed across the two age groups, indicating that the effect of dietary zinc status on the microbiome depends upon the context of host age. Many of the taxa that demonstrated these trends have previously been linked to aging or inflammatory phenotypes, and several of these taxa also associate with markers of inflammation in our study. For example, discrete clades of Lachnospiraceae responded to zinc status differently in aged mice versus young mice, and prior work has linked changes in Lachnospiraceae to mammalian host aging, especially in humans [54, 55]. Previous work has also found significant changes in Lachnospiraceae when hosts (*Gallus gallus*) are fed a zinc supplemented diet, and found unclassified Lachnospiraceae useful as a biomarker for characterizing host zinc status [56, 57]. Additionally, clades within the *Acetitafactor* genus are sensitive to dietary zinc, in an age-dependent manner, and also positively correlate with markers of inflammation, which are observations that align with previous reports with respect to modulations due to changes in dietary micronutrients and colonic inflammation [58, 59]. Furthermore, we observed an increase in *Lactobacillus* clades in response to zinc supplementation in old mice, but not young mice, and prior work has found that some species of *Lactobacillus* elicit immunomodulatory effects that improve immune response in the host [60, 61]. Additionally, a subset of Ruminococcaceae spp. positively associated with zinc deficiency in old mice, and positively associated with inflammation in old mice. A separate subset of Ruminococcaceae was negatively associated with inflammation in young mice, consistent

with previous work examining inflammation in humans [62]. These cladal differences within the Ruminococcaceae taxonomic group reinforces the need to examine the empirically derived phylogenetic groups within a microbiome dataset. Further, the results underscore the potential difficulties when comparing data between studies that may have had different types of taxonomic agglomeration.

While these associations cannot reveal cause and effect, the consistent link between these taxa, aging, and in some cases the age-related phenotype of inflammation suggests that these taxa may mediate how changes in dietary zinc status impact age-related inflammation, a hypothesis that future work should explore. Moreover, the fact that these associations are observed at the level of phylogenetic clades and not ASVs indicates that dietary zinc status does not select for specific strains per se, but rather more diverse phylogenetic lineage that are related through common descent, possibly because these lineages possess conserved genomic functions that underlie their relationship with zinc. Regardless, our study's multifactor design ultimately enabled our discovery of these trends which collectively underscore the importance of considering the interaction between factors that drive diversification of the gut microbiome.

## The effect of aging on the mouse gut microbiome is robust to study effects

A major question that complicates the interpretation of microbiome research results and discovery is whether patterns observed in an investigation are ultimately robust to experimental or study effects. These effects, which largely amount to sources of technical variation, result from experimental features that are often not explicitly controlled for, such as the impact of the research facility and animal cohort, and which may impact how the microbiome responds to the experimental variables being explored (e.g., exposure to dietary zinc) [63]. While these effects are typically uniform across the individuals subject to a specific investigation, efforts to integrate results across studies are confounded by these effects and it is frequently unclear if the conclusions drawn in a single study are ultimately robust to these effects [64, 65]. In this study, we applied a design that enabled quantitative assessment of study effects, in such that we purchased aged mice from multiple vendors and replicated the age component of our study across two different experiments. We generally find that study effects elicit minimal impact on the gut microbiome, at least with respect to how age affects gut microbiome diversification. In particular, we did not find any significant differences in overall beta diversity between the two studies when using a PERMANOVA test. When we restricted the analysis to comparisons of intra-age group beta dispersion, we did find a difference. These results indicate that, at least with respect to age effects, the overall effects observed in a mouse study tend to be relatively robust.

That said, one limitation to our investigation is that across both arms of the study, only a single dietary zinc experimental group is present. This limitation to our design resulted from the fact that suppliers of aged mice can only provide investigative teams with a limited number of aged individuals at a time due to resource scarcity. To ensure a consistent control for age while maintaining a sufficient sample size to facilitate discovery, we adopted the study design described here in light of this resource limitation. Regardless, study and diet effects are completely confounded, and consequently we are unable to assess whether the results of dietary micronutrient state on the gut microbiome are robust.

## Overall conclusions

We have shown that age is a predominant factor in determining the gut microbial composition in a mouse model. We found that over three-fourths of all genera had significantly differential abundance between the young and aged mice. Because of this large-scale shift in composition,

identifying specific gut microbiome components that correlate with either modulation of dietary zinc status or markers of host inflammation is a challenging task: the age status of an individual so substantially redefines the context of microbiome biodiversity that dietary zinc effects are unlikely to be consistent across age groups. This observation underscores the importance of considering multifactor designs when exploring microbiome sensitivity to endogenous and exogenous agents, as do our results age-specific effects of zinc status on microbial taxa. In total, these results indicate a complex interplay between host age, inflammation state, diet, and the gut microbiome.

## Supporting information

**S1 Checklist. ARRIVE guidelines 2.0 author checklist.**
(PDF)

**S1 Fig. Significant differences in dispersion between studies are observed when data are analyzed within and not between age groups.** Robust principal component analysis results are shown for combined ages, young mice and old mice, respectively (A-C). The intra-study dispersions are shown using boxplots (D-F) for combined ages, young mice, and old mice, respectively, with the first, median, and third quartiles shown by the hinges, and the whiskers extend to no more than 1.5 x inter-quartile range, with outliers plotted individually.
(TIF)

**S1 Table. Wilcoxon Rank Sum tests on genus abundance indicating 150 of 186 are significantly different between old and young mice.**
(XLSX)

**S2 Table. Wilcoxon Rank Sum tests confirm differential abundance of taxa at every taxonomic level (phylum through genus level).**
(XLSX)

**S3 Table. Generalized linear modeling results indicating 240 of 360 clades are significantly different between young and old mice.**
(XLSX)

**S4 Table. Generalized linear modeling results indicating 39 clades are significantly different between mice fed different levels of dietary zinc.**
(XLSX)

**S5 Table. Generalized linear modeling results indicating 103 clades are significantly associated with IL-6.**
(XLSX)

**S6 Table. Sample metadata including host serum zinc and whole blood LPS-induced IL-6 values.**
(XLSX)

## Acknowledgments

We thank the Oregon State University Center for Qualitative Life Sciences for their support in generating the DNA sequence data used in this study.

## Author Contributions

**Conceptualization:** Thomas J. Sharpton, Emily Ho.

**Data curation:** Edward W. Davis, II, Carmen P. Wong.

**Formal analysis:** Edward W. Davis, II, Holly K. Arnold, Kristin Kasschau, Thomas J. Sharpton.

**Funding acquisition:** Thomas J. Sharpton, Emily Ho.

**Investigation:** Carmen P. Wong.

**Methodology:** Edward W. Davis, II, Holly K. Arnold, Christopher A. Gaulke, Thomas J. Sharpton.

**Project administration:** Thomas J. Sharpton, Emily Ho.

**Resources:** Carmen P. Wong.

**Software:** Edward W. Davis, II, Holly K. Arnold, Christopher A. Gaulke.

**Supervision:** Thomas J. Sharpton, Emily Ho.

**Visualization:** Edward W. Davis, II, Holly K. Arnold.

**Writing – original draft:** Edward W. Davis, II, Carmen P. Wong, Holly K. Arnold, Thomas J. Sharpton.

**Writing – review & editing:** Edward W. Davis, II, Carmen P. Wong, Holly K. Arnold, Kristin Kasschau, Christopher A. Gaulke, Thomas J. Sharpton, Emily Ho.

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
