## [Decision Letter · Decision Letter 0]

13 Oct 2022

PONE-D-22-25498Age and micronutrient effects on the microbiome in a mouse model of zinc depletion and supplementationPLOS ONE

Dear Dr. Ho,

Thank you for submitting your manuscript to PLOS ONE. After careful consideration, we feel that it has merit but does not fully meet PLOS ONE’s publication criteria as it currently stands. Therefore, we invite you to submit a revised version of the manuscript that addresses the points raised during the review process.

Both reviewers felt the work was interesting but noted a number of issues that need to be more adequately addressed. In particular, the discussion would benefit from more information regarding predicted microbial metabolic changes in response to age/dietary zinc and a more thorough explanation of the connection to IL6 and existing work in the field.

We look forward to receiving your revised manuscript.

Kind regards,

Brenda A Wilson, Ph.D.

Academic Editor

PLOS ONE

Journal Requirements:

2. As part of your revision, please complete and submit a copy of the Full ARRIVE 2.0 Guidelines checklist, a document that aims to improve experimental reporting and reproducibility of animal studies for purposes of post-publication data analysis and reproducibility: https://arriveguidelines.org/sites/arrive/files/documents/Author%20Checklist%20-%20Full.pdf (PDF). 

Please include your completed checklist as a Supporting Information file. Note that if your paper is accepted for publication, this checklist will be published as part of your article.

3. To comply with PLOS ONE submissions requirements, in your Methods section, please provide additional information regarding the experiments involving animals and ensure you have included details on (1) methods of sacrifice, (2) methods of anesthesia and/or analgesia, and (3) efforts to alleviate suffering.

Reviewers' comments:

Reviewer's Responses to Questions

**Comments to the Author**

1. Is the manuscript technically sound, and do the data support the conclusions?

Reviewer #1: Yes

Reviewer #2: Partly

2. Has the statistical analysis been performed appropriately and rigorously? 

Reviewer #1: Yes

Reviewer #2: Yes

3. Have the authors made all data underlying the findings in their manuscript fully available?

Reviewer #1: Yes

Reviewer #2: Yes

4. Is the manuscript presented in an intelligible fashion and written in standard English?

Reviewer #1: Yes

Reviewer #2: Yes

5. Review Comments to the Author

Reviewer #1: The paper was overall quite good and the results weren't unexpected based on other mouse/zinc/microbiome studies, but I would have liked to see data on changes in predicted microbial metabolic pathways of the bacteria (i.e. KEGG analysis or something) with changes in age/dietary zinc.

Line 88

To further support the importance of considering zinc deficiency/bioavailability in combination with another condition, suggest mentioning this reference on zinc and pregnancy effects on the microbiome: DOI: 10.3389/fnins.2019.01295

Line 109-110

Could the authors please further clarify this statement? How does the microbiome contribute to changes in zinc availability? Does microbial metabolite profile change with host age?

Line 125

Could the authors add a sentence that summarizes the key points of their recently published work (reference 19)?

Line 161

The authors use “IL-6” in the methods section, and then use “IL6” in the discussion – suggest sticking with one abbreviation for consistency. Also, could the author define IL6 as Interleukin 6 in the first mention of the abbreviation?

Line 168

Could the authors include either tables or figures in either the main text or supplemental information that show the results of measured serum zinc and IL6 (i.e. show if there are differences in serum zinc and or IL6 between the ZA/ZD and ZA/ZS groups)?

Line 298

For Figure 1D, could it be clarified in the manuscript in the Figure caption if the subjects shown are on a specific diet? Is there a way to show the average of specific groups - i.e. average all the aged mice on the zinc adequate diet instead of showing all subjects in each group?

Line 536

This reference mentions that Lachnospiraceae can potentially be predictive of zinc status, which may further support the authors’ statement that this bacteria family responded to changes in dietary zinc: DOI: 10.3390/nu13103399

Line 560

The authors did excellent work looking at which bacteria were responsive to changes in dietary zinc. The authors could consider looking into predicted functions of the bacteria based on the bacteria present to help strengthen the paper. What are the specific (predicted) functional capacities that are impacted by dietary zinc? Do the authors have an idea which conserved genomic functions are related to dietary zinc?

Reviewer #2: In this work, Davis et al. used a multifactor approach to describe the complex interplay between age and nutritional factors in the gut microbiota. They determine that the microbial composition is impacted strongly by age (young v. old) but less so by nutrient zinc. Though the overall composition was not changed by zinc, they did detect alterations in key phylogenetic clades that were zinc dependent. This work was interesting but would benefit from a more thorough explanation of the connection to IL6 and existing work in the field. Elaboration on some points listed below would greatly help with comprehension.

Key points:

- There is a disconnect between the primary story and the introduction of IL6. Inflammation is minimally mentioned and ‘inflammaging’ introduced only in the discussion.

o Ln 158-162. Is this how IL6 was measured? If so, please explicitly state IL6 measured from blood.

o Sentence at Ln 167-168 is confusing. Was LPS-induced IL6 included in this study?

- This manuscript needs more references. The discussion reads as a reiteration of the conclusions and many topics are not well developed/supported with citations. Authors should outline novelty from these findings and how they fit into the current understanding of the gut microbiome. Consider including information from the human microbiome literature.

- Dispersion differences across experiments are not well addressed. Ln 286-292. Do the authors believe this to be diet dependent? This would be helpful to know before combining all data together for models.

o I appreciate the authors purchasing mice from different locations. Was baseline sequencing performed? Could this contribute to dispersion differences?

- Fig 1. Adequate and deficient/supplemented mice should be made clear by color or symbol. I would recommend replacing fig 1C with supp fig 1A.

- Understanding directionality of differences is confusing throughout. Adding abundance levels of each organism to supplemental tables would be helpful to know when abundance is increasing/decreasing across age or diet. This is clear in fig 2.

- Please include information about sample size, type (fecal, cecum, colon), and timepoint in methods and figure legends.

Minor:

- Please check usage throughout for microbiota v. microbiome.

- Ln 34. Please remove the word ‘elevated’.

- Ln 47. Please remove ‘at each’.

- Ln 49. Replace ‘components’ with ‘taxa’

- Ln 52-53. Remove ‘of microbiota’

- Ln 54-55. Remove ‘there exists’ and change to ‘interplay exists’

- Ln 74. Remove first ‘more’

- Ln 103. Remove ‘,including zinc,’ this is redundant.

- Ln 332-333. Remove ‘in order to detect these taxa’

- Ln 341-344. Remove this section. This is restated in the following section

- Ln 369-371. References? These have a different format (#) v. [#].

- It would be helpful to have panel titles added to figures 3 and 4. Or changing symbols so age and diet are clearly different.

6. PLOS authors have the option to publish the peer review history of their article (what does this mean?). If published, this will include your full peer review and any attached files.

Reviewer #1: No

Reviewer #2: No

---

## [Author Response · Author response to Decision Letter 0]

25 Nov 2022

We appreciate the reviewers’ comments and suggestions to improve our manuscript. We have provided clarifications and/or changes to our revised manuscript as detailed below. We hope these clarifications and corrections fully address the reviewer’s concerns.

We followed these templates with the first submission and again with the revision.

2. As part of your revision, please complete and submit a copy of the Full ARRIVE 2.0 Guidelines checklist, a document that aims to improve experimental reporting and reproducibility of animal studies for purposes of post-publication data analysis and reproducibility: https://arriveguidelines.org/sites/arrive/files/documents/Author%20Checklist%20-%20Full.pdf (PDF). 

Please include your completed checklist as a Supporting Information file. Note that if your paper is accepted for publication, this checklist will be published as part of your article.

We have filled out the ARRIVE guidelines checklist and modified the text as appropriate to conform to the checklist. The checklist is included as supporting information S1 Checklist.

3. To comply with PLOS ONE submissions requirements, in your Methods section, please provide additional information regarding the experiments involving animals and ensure you have included details on (1) methods of sacrifice, (2) methods of anesthesia and/or analgesia, and (3) efforts to alleviate suffering.

We have provided the method of sacrifice, care during the experiment, and that our care was approved by the Oregon State University (OSU) Institutional Laboratory Animal Care and Use Committee, and adheres to the international standards of animal care as established by the Association for Assessment and Accreditation of Laboratory Animal Care International.

Accession numbers are included, and data are now available on the web. https://www.ncbi.nlm.nih.gov/bioproject/PRJNA831825

5. Review Comments to the Author

Reviewer #1: The paper was overall quite good and the results weren't unexpected based on other mouse/zinc/microbiome studies, but I would have liked to see data on changes in predicted microbial metabolic pathways of the bacteria (i.e. KEGG analysis or something) with changes in age/dietary zinc.

Line 88

To further support the importance of considering zinc deficiency/bioavailability in combination with another condition, suggest mentioning this reference on zinc and pregnancy effects on the microbiome: DOI: 10.3389/fnins.2019.01295

Thank you for sending us this reference. We have included it in the introduction as a link between zinc status and microbiome.

Line 109-110

Could the authors please further clarify this statement? How does the microbiome contribute to changes in zinc availability? Does microbial metabolite profile change with host age?

Previous work has shown that the gut microbiome changes with age. We hypothesize that these changes in the gut microbiome might affect zinc availability to the host as the zinc requirements of the microbes change over time. We tested this hypothesis in the current study. We are unable to answer the question regarding microbial metabolite profiles as we did not measure the metabolites in the current study. This is an important area for future study.

Line 125

Could the authors add a sentence that summarizes the key points of their recently published work (reference 19)?

Thank you for the suggestion. We added statements regarding the previously published conclusions of the host metabolic and inflammation marker analysis.

Line 161

The authors use “IL-6” in the methods section, and then use “IL6” in the discussion – suggest sticking with one abbreviation for consistency. Also, could the author define IL6 as Interleukin 6 in the first mention of the abbreviation?

Thank you for mentioning this inconsistency. We now define IL-6 in the introduction and use IL-6 rather than IL6 throughout the manuscript.

Line 168

Could the authors include either tables or figures in either the main text or supplemental information that show the results of measured serum zinc and IL6 (i.e. show if there are differences in serum zinc and or IL6 between the ZA/ZD and ZA/ZS groups)?

We’ve included the zinc metadata in a table (S6 table) along with the other sample data. We have previously published the host metabolic data, which we are not replicating in the current analysis of microbiome data.

Line 298

For Figure 1D, could it be clarified in the manuscript in the Figure caption if the subjects shown are on a specific diet? Is there a way to show the average of specific groups - i.e. average all the aged mice on the zinc adequate diet instead of showing all subjects in each group?

Our desire in Figure 1 is to show the differences between the age groups regardless of diet. We intentionally did not label diet to encourage looking at age only in this part of the analysis. This is necessary to conclude that it is possible to combine the data from the deficiency and supplementation studies.

Effectively, the analysis must occur in a two step process, one step being age, and the second being diet, so we walk the reader through those steps as well. We use the stacked barplots to highlight the differences based on age, with Bacteroides vs Parabacteroides being the most prominent. After we establish that combining the two studies is possible, we then analyze the age by diet groups for the remainder of the manuscript, starting with the genus level boxplots in Figure 2.

Line 536

This reference mentions that Lachnospiraceae can potentially be predictive of zinc status, which may further support the authors’ statement that this bacteria family responded to changes in dietary zinc: DOI: 10.3390/nu13103399

Thank you. We added two additional citations regarding Lachnospiraceae in the broiler chicken gut microbiome research.

Line 560

The authors did excellent work looking at which bacteria were responsive to changes in dietary zinc. The authors could consider looking into predicted functions of the bacteria based on the bacteria present to help strengthen the paper. What are the specific (predicted) functional capacities that are impacted by dietary zinc? Do the authors have an idea which conserved genomic functions are related to dietary zinc?

We elected not to impute functions using a tool, e.g. picrust2, as for non-human systems, picrust2 is significantly more inaccurate - https://doi.org/10.1186/s40168-020-00815-y. We would suggest that shotgun metagenomic work could examine this question in the future.

Reviewer #2: In this work, Davis et al. used a multifactor approach to describe the complex interplay between age and nutritional factors in the gut microbiota. They determine that the microbial composition is impacted strongly by age (young v. old) but less so by nutrient zinc. Though the overall composition was not changed by zinc, they did detect alterations in key phylogenetic clades that were zinc dependent. This work was interesting but would benefit from a more thorough explanation of the connection to IL6 and existing work in the field. Elaboration on some points listed below would greatly help with comprehension.

Key points:

- There is a disconnect between the primary story and the introduction of IL6. Inflammation is minimally mentioned and ‘inflammaging’ introduced only in the discussion.

While we introduce the concept of age-related inflammation in the introduction, we agree that we focused more on zinc status and age in terms of interaction with the gut microbiome. We have explicitly added mention of inflammation along with the age and diet now throughout the manuscript.

o Ln 158-162. Is this how IL6 was measured? If so, please explicitly state IL6 measured from blood.

Thank you for mentioning this oversight. We have expanded the methods section to more thoroughly explain how the IL-6 measurements were done, rather than rely on readers to follow the cited reference.

o Sentence at Ln 167-168 is confusing. Was LPS-induced IL6 included in this study?

Yes, and we have clarified this statement as above.

- This manuscript needs more references. The discussion reads as a reiteration of the conclusions and many topics are not well developed/supported with citations. Authors should outline novelty from these findings and how they fit into the current understanding of the gut microbiome. Consider including information from the human microbiome literature.

We added references to the discussion that we previously missed, thank you for noticing those omissions.

One of our described conclusions, that age is a primary driver of gut microbiome composition, is consistent with what has been seen in human studies, that are cited, and is not always directly controlled for in mouse studies. Further, we hypothesize why our diet effects are not as large as in other dietary interventions, as we only modulate a single micronutrient, and we cite those other studies, including those in humans. We also discuss specific taxa that change in our study and were also shown to change in previous studies, including human studies, that we cite.

With the added citations, we believe that the current study has been presented within the current understanding of the gut microbiome within the lens of aging, zinc status, and inflammation.

- Dispersion differences across experiments are not well addressed. Ln 286-292. Do the authors believe this to be diet dependent? This would be helpful to know before combining all data together for models.

We unfortunately cannot determine if the dispersion effects between the two studies are diet dependent because diet and study are completely confounded (mentioned on lines 593-595). Despite this limitation, we are able to compare the control mice between the two studies and indeed see the differences in dispersion when comparing mice of the same age group (lines 286-289). Therefore, the most parsimonious explanation was that the differences in dispersion were due to study effect rather than diet effect (because control mice were different). Additionally, there were no significant differences in dispersion between diet groups in the same study. We did see a significant difference between age groups, however (lines 283-286).

We later controlled for this effect by running four models, not a single combined model, one for each age and diet/study group.

o I appreciate the authors purchasing mice from different locations. Was baseline sequencing performed? Could this contribute to dispersion differences?

We attempted to control for this, and did measure a week 0 timepoint, which was after a 1 month acclimation period (lines 135-138). All young mice were from Jackson labs, but from different rooms. Studies were not done concurrently due to being unable to source the aged mice all at the same time from NIA. Mice purchased at the same time, regardless of origin, grouped according to beta diversity analysis and PERMANOVA (Fig S1).

- Fig 1. Adequate and deficient/supplemented mice should be made clear by color or symbol. I would recommend replacing fig 1C with supp fig 1A.

Thank you for the suggestion. However, the purpose of Figure 1 is to demonstrate that grouping mice by age, regardless of diet/study, is statistically valid. Therefore, including diet would confuse the figure and the reader. We include those data in Fig S1 to show that we thoroughly examined dispersion differences between the mice.

- Understanding directionality of differences is confusing throughout. Adding abundance levels of each organism to supplemental tables would be helpful to know when abundance is increasing/decreasing across age or diet. This is clear in fig 2.

Understood. Text was added to the column headers to explain the meaning of the model coefficients.

- Please include information about sample size, type (fecal, cecum, colon), and timepoint in methods and figure legends.

We have added sample size in the methods, a supp table with the metadata, and sample size values in the figures and/or legends where appropriate.

Minor:

We have implemented these changes. Thank you.

- Please check usage throughout for microbiota v. microbiome.

- Ln 34. Please remove the word ‘elevated’.

- Ln 47. Please remove ‘at each’.

- Ln 49. Replace ‘components’ with ‘taxa’

- Ln 52-53. Remove ‘of microbiota’

- Ln 54-55. Remove ‘there exists’ and change to ‘interplay exists’

- Ln 74. Remove first ‘more’

- Ln 103. Remove ‘,including zinc,’ this is redundant.

- Ln 332-333. Remove ‘in order to detect these taxa’

- Ln 341-344. Remove this section. This is restated in the following section

- Ln 369-371. References? These have a different format (#) v. [#].

Sorry for the confusion. We mention that the numbers in parentheses are counts of number of clades per group in line 367.

- It would be helpful to have panel titles added to figures 3 and 4. Or changing symbols so age and diet are clearly different.

Panel titles were added in Fig 3 and 4.

---

## [Decision Letter · Decision Letter 1]

7 Dec 2022

Age and micronutrient effects on the microbiome in a mouse model of zinc depletion and supplementation

PONE-D-22-25498R1

Dear Dr. Ho,

We’re pleased to inform you that your manuscript has been judged scientifically suitable for publication and will be formally accepted for publication once it meets all outstanding technical requirements.

Kind regards,

Brenda A Wilson, Ph.D.

Academic Editor

PLOS ONE

Additional Editor Comments (optional):

Reviewers' comments:

Reviewer's Responses to Questions

**Comments to the Author**

1. If the authors have adequately addressed your comments raised in a previous round of review and you feel that this manuscript is now acceptable for publication, you may indicate that here to bypass the “Comments to the Author” section, enter your conflict of interest statement in the “Confidential to Editor” section, and submit your "Accept" recommendation.

Reviewer #1: All comments have been addressed

Reviewer #2: All comments have been addressed

2. Is the manuscript technically sound, and do the data support the conclusions?

Reviewer #1: Yes

Reviewer #2: Yes

3. Has the statistical analysis been performed appropriately and rigorously? 

Reviewer #1: Yes

Reviewer #2: Yes

4. Have the authors made all data underlying the findings in their manuscript fully available?

Reviewer #1: Yes

Reviewer #2: Yes

5. Is the manuscript presented in an intelligible fashion and written in standard English?

Reviewer #1: Yes

Reviewer #2: Yes

6. Review Comments to the Author

Reviewer #1: The authors have appropriately addressed the comments provided by this reviewer. The manuscript is now acceptable.

Reviewer #2: I appreciate the efforts the authors have made to make this manuscript more cohesive, all of my concerns were addressed.

7. PLOS authors have the option to publish the peer review history of their article (what does this mean?). If published, this will include your full peer review and any attached files.

Reviewer #1: No

Reviewer #2: No

---

## [Editor Report · Acceptance letter]

12 Dec 2022

PONE-D-22-25498R1 

Age and micronutrient effects on the microbiome in a mouse model of zinc depletion and supplementation 

Dear Dr. Ho:

I'm pleased to inform you that your manuscript has been deemed suitable for publication in PLOS ONE. Congratulations! Your manuscript is now with our production department. 

Kind regards, 

on behalf of

Dr. Brenda A Wilson 

Academic Editor

PLOS ONE